# The Worm-Specific Immune Response in Multiple Sclerosis Patients Receiving Controlled *Trichuris suis* Ova Immunotherapy

**DOI:** 10.3390/life11020101

**Published:** 2021-01-29

**Authors:** Ivet A. Yordanova, Friederike Ebner, Axel Ronald Schulz, Svenja Steinfelder, Berit Rosche, Anna Bolze, Friedemann Paul, Henrik E. Mei, Susanne Hartmann

**Affiliations:** 1Institute of Immunology, Center for Infection Medicine, Freie Universität Berlin, D-14163 Berlin, Germany; ivet.yordanova@fu-berlin.de (I.A.Y.); friederike.ebner@fu-berlin.de (F.E.); 2German Rheumatism Research Center Berlin (DRFZ), a Leibniz Institute, D-10117 Berlin, Germany; axel.schulz@drfz.de (A.R.S.); mei@drfz.de (H.E.M.); 3Max Delbrück Center for Molecular Medicine, D-13125 Berlin, Germany; svenja.steinfelder@mdc-berlin.de; 4Department of Neurology and Experimental Neurology, Charité—Universitätsmedizin Berlin, D-10117 Berlin, Germany; berit.rosche@gmx.de; 5Clinical and Experimental Multiple Sclerosis Research Center, Charité—Universitätsmedizin Berlin, D-10117 Berlin, Germany; friedemann.paul@charite.de; 6NeuroCure Clinical Research Center, Charité—Universitätsmedizin Berlin, D-10117 Berlin, Germany; af.bolze@yahoo.com; 7Experimental and Clinical Research Center, Max Delbrück Center for Molecular Medicine, Charité—Universitätsmedizin Berlin, D-10117 Berlin, Germany

**Keywords:** *Trichuris suis*, TSO, helminth therapy, multiple sclerosis, mass cytometry, clinical trial

## Abstract

Considering their potent immunomodulatory properties, therapeutic applications of *Trichuris suis* ova (TSO) are studied as potential alternative treatment of autoimmune disorders like multiple sclerosis (MS), rheumatoid arthritis (RA), or inflammatory bowel disease (IBD). Clinical phase 1 and 2 studies have demonstrated TSO treatment to be safe and well tolerated in MS patients, however, they reported only modest clinical efficacy. We therefore addressed the cellular and humoral immune responses directed against parasite antigens in individual MS patients receiving controlled TSO treatment (2500 TSO p.o. every 2 weeks for 12 month). Peripheral blood mononuclear cells (PBMC) of MS patients treated with TSO (*n* = 5) or placebo (*n* = 6) were analyzed. A continuous increase of serum IgG and IgE antibodies specific for *T. suis* excretory/secretory antigens was observed up to 12 months post-treatment. This was consistent with mass cytometry analysis identifying an increase of activated HLA-DR^high^ plasmablast frequencies in TSO-treated patients. While stable and comparable frequencies of total CD4^+^ and CD8^+^ T cells were detected in placebo and TSO-treated patients over time, we observed an increase of activated HLA-DR^+^CD4^+^ T cells in TSO-treated patients only. Frequencies of Gata3^+^ Th2 cells and Th1/Th2 ratios remained stable during TSO treatment, while Foxp3^+^ Treg frequencies varied greatly between individuals. Using a *T. suis* antigen-specific T cell expansion assay, we also detected patient-to-patient variation of antigen-specific T cell recall responses and cytokine production. In summary, MS patients receiving TSO treatment established a *T. suis*-specific T- and B-cell response, however, with varying degrees of T cell responses and cellular functionality across individuals, which might account for the overall miscellaneous clinical efficacy in the studied patients.

## 1. Introduction

More than a decade ago the innovative idea arose to use parasitic helminths with active immunomodulatory properties to treat autoimmune and inflammatory diseases. Considering the potential benefits and risks of applying a known infectious organism to patients suffering from inflammatory conditions, the porcine whipworm *Trichuris suis* was preselected as a promising candidate for treatment [1]. *T. suis* is the species most closely related to the human-infecting parasite *T. trichiura* and oral administration of *T. suis* eggs (*T. suis* ova, TSO) typically results in a self-limiting colonization of the human gut [2,3]. In contrast to ascarids or hookworms, whipworms like *T. suis* have a direct, strictly enteric life cycle. Following ingestion of infective eggs (TSO) containing stage 1 larvae (L1), these larvae hatch and invade the mucosa of the large intestine of the host. There, *Trichuris* larvae undergo four moulting steps. As they reach the adult stage, the thinner anterior end of the worm is found burrowed into the epithelial cell layer, while the thicker posterior end protrudes freely into the intestinal lumen [4]. In humans, the pig whipworm *T. suis* typically does not develop into sexually mature adults. Thus, in order to maintain hatched *T. suis* larvae in the non-natural human host, repeated treatments with TSO are required.

The potential of TSO-based therapies to treat autoimmune diseases has been tested in a variety of clinical studies (reviewed in [5]) with some promising, but also various disappointing clinical outcomes. Notably, such alternative helminth-based treatment has received considerable attention in recent years as a potentially useful therapy for treatment of multiple sclerosis (MS). MS is a debilitating, demyelinating disease of the central nervous system, most commonly presented as relapsing-remitting multiple sclerosis (RRMS), characterized by transient episodes of neurological dysfunction, mostly in young females [6,7]. Similar to other autoimmune diseases, the inverse correlation of natural helminth infections and the prevalence of MS in helminth-endemic countries was an initial driver of the idea of helminth-based therapy as a novel treatment option [8]. Correale and Farez have demonstrated that the administration of anthelmintic drugs led to an increase of MS disease activity, suggestive of active suppression of MS mediated by helminth parasites [9]. Moreover, several studies using experimental autoimmune encephalomyelitis (EAE), the murine model for MS, have tested infective helminth eggs, larvae or the application of helminth-derived molecules in both prophylactic and therapeutic treatment settings, reporting an improved neuroinflammatory disease outcome (reviewed in [10]).

Until now, four clinical phase 1 and phase 2 trials investigating treatment effects of either oral inoculation of TSO or intradermal application of *Necator americanus* (hookworm) larvae in MS patients have been completed [11,12,13,14]. While all studies demonstrate the safety of controlled administration of both *T. suis* and *N. americanus*, none of them reported strong therapeutic efficacy, expressed in radiological or clinical disease activity [11,12,13,14]. Immunological results of the HINT-2 study demonstrated that TSO given at a dose of 2500 TSO every 2 weeks over 10 months resulted in mild eosinophilia and *T. suis*-specific antibodies [13]. Additionally, the study showed an increase of CD4^+^CD127^−^CD25^high^ regulatory T cells (Treg) in the circulation of TSO-treated patients, compared with placebo-treated MS patients. Live *T. suis* larvae were also observed in one of two TSO-treated patients undergoing clinically indicated colonoscopy, indicating that *T. suis* larvae colonized the host intestine and induced a specific immune response [13]. Similarly, preliminary results from four patients with secondary progressive MS receiving TSO treatment for compassionate use indicated a slight downregulation of Th1-associated cytokines [15]. 

Combined, these results prompted us to set up a small monocentric, prospective, randomized, placebo-controlled, double-blind, phase II study enrolling patients with RRMS and clinically isolated syndrome (CIS) to investigate in greater detail changes in different immune parameters, as well as the individual helminth-specific T cell signatures of MS patients receiving prolonged TSO treatment (TRIOMS) [16].

## 2. Materials and Methods

### 2.1. Patient Recruitment, Study Design, and Treatment Allocation

Eleven patients diagnosed with relapsing-remitting multiple sclerosis (RRMS) according to the revised McDonald criteria [17] (CIS) with clinical activity were recruited for experimental treatment with *T. suis* ova (TRIOMS). This study was a monocentric, prospective, randomized, placebo-controlled, double-blind, phase II pilot study, conducted at the Department of Neurology and the NeuroCure Clinical Research Center of the Charité-Universitätsmedizin Berlin [16] (ClinicalTrials.gov: NCT01413243). The study was approved by the local ethics committee and the German Federal Institute for Drugs and Medical Devices (BfArM).

Individuals were recruited between September 2012 and March 2015. Patients aged between 18 and 65 years, with a score of ≤4.0 on the Expanded Disability Status Scale (EDSS) [18], with disease activity on brain MRI, and without any assigned standard immunomodulatory therapies for at least 3 months prior to the study, were randomized 1:1 to receive orally either 2500 *Trichuris suis* ova (TSO) or placebo every 2 weeks for 12 months (Figure 1a). The TSO suspension (sterile phosphate buffered saline, PBS) and placebo aliquots were supplied by OVAMED GmbH (Barsbüttel, Germany).

Patient visits were carried out after 3, 6, 9, and 12 months after the start of TRIOMS study (V1-V4; Figure 1a). The primary clinical endpoint was the cumulative number of new hyperintense lesions identified on T2-weighted brain MRI during the treatment period of 12 months. Apart from treatment safety and tolerability, other secondary clinical endpoints included the annualized relapse rate and the proportion of relapse-free patients. In vivo mechanisms of action of TSO in the patients were assessed by extensive laboratory and immunological investigations. Due to low patient compliance with regular attendance of medical examinations and blood sampling, as well as limited sample/cell quantity collected at several sampling timepoints, not all immunological assays and clinical examinations could be performed for all patients and all timepoints of the study. Therefore, for serological analysis, immunoblotting, flow cytometry, and in vitro expansion and restimulation samples were available from four of five TSO-treated (002, 008, 010, 013) and four of six placebo-treated patients (003, 006, 011, 012) only. For mass cytometry, samples were available from four TSO-treated (002, 005, 008, 010) and six placebo-treated patients (001, 003, 004, 006, 011, 012). Due to limited patient sample size, we designed the investigation as a pilot study employing detailed immune-centered exploratory analysis at the individual patient level.

### 2.2. Peripheral Blood Mononuclear Cell (PBMC) Isolation

Briefly, 20–40 mL of peripheral blood were collected per visit in heparin tubes. Peripheral blood mononuclear cells (PBMC) were isolated via density gradient centrifugation over human Pancoll (1.077 g/mL, PAN-Biotech, Aidenbach, Germany). PBMC aliquots were cryopreserved at −80 °C in RPMI-1640 supplemented with 10% DMSO and 10% FCS until further use.

### 2.3. Mass Cytometry and Data Analysis

PBS and CyTOF staining medium (CSM) used in mass cytometric assays were prepared as described before [19]. All antibodies and metal isotopes and their sources are listed in Appendix A. In-house antibody conjugations with lanthanide and indium isotopes were carried out using MAXPAR X8 labeling kits (Fluidigm, Markham, ON, Canada) according to the manufacturer’s instructions. Platinum or palladium labeling was performed as described [20,21]. Cisplatins carrying isotopically enriched Pt were purchased from or kindly provided by Fluidigm. Cell events were detected using an iridium-containing DNA intercalator (Fluidigm). m-DOTA-Rh, used for discrimination of dead cells, was prepared from DOTA-maleimide (Macrocyclics, Dallas, TX, USA) and rhodium chloride (Sigma-Aldrich) as described [22] and stored at 4 °C. Antibody–metal conjugates were used at concentrations optimized to separate cell populations expressing or lacking a particular antigen. All antibody cocktails were prepared in advance and cryopreserved at −80 °C [19]. 

Cryopreserved PBMC were thawed as described [19] and cell-surface-barcoded using a set of palladium- and platinum-isotope-labeled β2-microglobulin (B2M) antibodies [20,21]. Samples were run in a total of two pools of barcoded samples, prepared and acquired 1 week apart. Longitudinal samples of a given donor were always acquired as parts of the same sample pool. For this, PBMC were incubated for 30 min at 4 °C with combinations of B2M conjugates in a 5-choose-2 barcode scheme. Next, the barcoded cells were washed four times with 1.2 mL CSM (500× *g*, 5 min, 4 °C), pooled and further processed together. Up to 17 × 10^6^ pooled PBMC were stained in a 200 µL reaction with antibody cocktail 1 (Supplem. Table 1) for 30 min at 4 °C and washed twice with 1 mL CSM (500× *g*, 5 min, 4 °C). This was followed by a second 200 µL staining reaction with antibody cocktail 2 (Supplem. Table 1) that included streptavidin-conjugated silver nanoparticles [23]. After an incubation of 30 min at 4 °C, the PBMC pool was washed once with 1 mL PBS (500× *g*, 5 min, 4 °C), and incubated in 1 mL 10 µM mDOTA-^103^Rh solution in PBS for 5 min at room temperature (RT) to label dead cells for their later exclusion. The sample volume was filled up to 1500 µL with CSM followed by cell pelleting and supernatant aspiration. Then, the cells were washed once with 1 mL CSM, once with 1 mL PBS, and finally resuspended in a 2% PFA solution (diluted from 16% stock with PBS, Electron Microscopy Sciences, Hatfield, PA) and incubated at 4 °C overnight. On the next day, 500 µL CSM were added, cells were pelleted (700× *g*, 5 min, 4 °C) and subsequently washed once with 1 mL CSM. Next, the sample pool was incubated for 25 min at room temperature (RT) in 500 µL 1 × permeabilization buffer (diluted with Millipore water from 10 × permeabilization buffer; Thermo Fisher Scientific, Waltham, MA, USA) supplemented 1:500 *v*/*v* with 0.125 µM iridium-based DNA intercalator. Cells were then washed with 1 mL CSM. Thereafter, samples were resuspended in 1 mL PBS and counted with a MACSQuant flow cytometer (Miltenyi Biotec, Bergisch-Gladbach, Germany). Prior to acquisition on the CyTOF instrument (Fluidigm), samples were washed twice with 1 mL Millipore water and pelleted by centrifugation at 800× *g*, 5 min, 4 °C. Cells were then resuspended in an appropriate volume of Millipore water to a maximum of 5 × 10^5^ cells/mL supplemented with EQ 4 element calibration beads (1:10, *v/v*) (Fluidigm) and filtered through 35 µm cell strainer cap tubes (BD, San Jose, CA, USA) prior to acquisition.

Mass cytometry was performed on CyTOF version 1 (operated as described before [23,24]) using instrument software version v6.0.626. The sample was acquired at a speed of 45 µL/min using the Supersampler injection system (Victorian Airship, Alamo, CA, USA). The instrument was run in dual calibration mode, with noise reduction turned on and event length thresholds set to 10 and 75. Raw data was converted to FCS 3.0 and normalized using EQ 4 element calibration beads and the CyTOF instrument software. Next, EQ 4 element calibration beads were excluded by gating on DNA(^193^Ir)^+^/^140^Ce^–^ cell events in FlowJo (Version 10.4, TreeStar, Ashland, OR, USA). After further removal of cell aggregates and ion cloud fusion events according to DNA and cell length parameters, the pooled sample was deconvoluted using Boolean combinations of manual gates, as previously described [25]. The resulting individual samples were further gated on CD45^+^/Dead discriminator(^103^Rh)^−^ cells and imported into the OMIQ.ai data analysis software (Santa Clara, CA, USA). An opt-SNE dimension reduction [26] was performed with the markers indicated in Appendix A and manually drawn gates on the tSNE1/tSNE2 projection were applied. Statistical analysis was performed using GraphPad Prism Version 8 (San Diego, CA, USA).

### 2.4. Antibody Detection by Enzyme-Linked Immunosorbent Assay (ELISA)

Total and *T. suis*-specific antibody titers in blood serum were quantified via sandwich ELISA. Briefly, 96-well flat-bottom Maxisorp plates (Thermo Fisher Scientific, Waltham, MA, USA) were coated with 5 µg/mL *T. suis* adult excretory/secretory antigen (TsAdES) and were incubated overnight at 4 °C. Plates were washed using a Tecan Hydrospeed microplate washer and blocked with 200 µL 3% BSA diluted in PBS for 1 h at room temperature before 50 µL per well of samples were added. Internal standards for each assay were generated using a pooled serum sample from TSO-treated patients 002, 010, and 013 (Visit 4) as a reference positive control. Two-fold serial dilutions were performed on the reference positive control sample, with a top dilution of 1:2500 for IgG and 1:10 for IgG1-4 and IgE detection. Plates were incubated with the samples and the pooled reference control dilution series for 2 h at room temperature, after which 50 µL of the respective detection antibodies were added for 1 h. For IgE quantification, following sample incubation in the IgG4-coated plate, samples and blanks were transferred to the IgE-coated plate (IgG4 depletion step). The samples were then incubated in the IgE plate for 2 h at room temperature. After washing, 50 µL of POX substrate (1 tablet tetramethylbenzidine dihydrochloride in 10 mL phosphate-citrate buffer supplemented with 5 µL H2O2) or alkaline phosphatase (AP) substrate (1 tablet phosphatase substrate in 10 mL carbonate buffer supplemented with 10 µL 1 M MgCl_2_) were added for 15 min (POX) or 30 min (AP) and plates were incubated either for 15 min at room temperature (POX) or for 30 min at 37 °C (AP). To stop the enzymatic reaction, 25 µL of 1 M H_2_SO_4_ (POX) or 100 mM EDTA (AP) was added to each well and the signals was measured on a Biotek Synergy H1 Hybrid Reader. The following dilutions were used for serum samples: 1:10,000 for IgG and 1:20 for IgG1-4 and IgE. HRP-conjugated human IgG, IgG1, IgG3 were from OriGene (ME, USA), HRP-conjugated IgG4 from GeneTex (CA, USA), and AP-conjugated human IgE from Biomol (Hamburg, Germany). Parasite-specific antibody levels in serum are reported here as Optical Density (OD) values.

### 2.5. Generation of Parasite Antigen Material

*Trichuris suis* parasite antigens used to study *Trichuris*-specific antibody responses in serum by ELISA and Western blotting, as well as for the expansion and restimulation of *Trichuris*-specific T cells in vitro were generated as follows: *T. suis* adult worms were manually collected from the caecum and colon of *T. suis*-infected pigs. Worms were washed four times for 15 min each in RPMI-1640 (PAN-Biotech, Aidenbach, Germany) supplemented with 200 U/mL penicillin, 200 µg/mL Streptomycin, 1.35 µg/mL Amphotericin B (all purchased from PAN-Biotech) by sedimentation and were incubated overnight in RPMI-1640 supplemented with 1% Glucose (Sigma-Aldrich, St. Louis, US), 100 U/mL Penicillin, 100 µg/mL Streptomycin and 0.625 µg/mL Amphotericin B. Whole adult worm lysate (TsAd) antigen was produced by snap freezing the parasites and homogenization using a mortar and pestle. The homogenized material was then dissolved in PBS, filter sterilized and stored at −20°C until further use. For the generation of *Trichuris* excretory-secretory (antigens (TsAdES), culture medium was replaced and worms were incubated for another 5 days, while conditioned media was collected every 2nd day, filter-sterilized (0.45 µm Minisart Syringe Filter, Sartorius AG, Göttingen, Germany) and stored at −20 °C until further use. Filter-sterilized conditioned media from both male and female *T. suis* was further concentrated using centrifugal concentrators (Vivaspin, 5 KDa cut off, Sartorius). For the generation of L1 larval antigen (TsL1), freshly hatched larvae were hatched from embryonated TSO and were snap frozen and homogenized. The material was then dissolved in PBS and stored at -20°C until further use.

### 2.6. Immunoblot Analysis

SDS-PAGE protein separation was performed using a 12% separating gel and a 6% stacking gel. A total of 15 µg of TsAdES antigen was applied per lane. Following separation, proteins were transferred to a nitrocellulose membrane (1 h at 80 mA). Membranes were blocked for 1 h at room temperature with 5% non-fat dry milk in TBS/0.2% Tween buffer. Following the blocking step, membranes were washed three times in TBS/0.2% Tween (TBST) and were incubated overnight on a shaker at 4 °C with serum samples (diluted 1:200 in TBST supplemented with 5% dry milk). After overnight incubation, membranes were washed three more times in TBST buffer and were then incubated for 1 h at room temperature with AP-conjugated goat anti-human IgG (diluted 1:2000 in TBST/5% dry milk) (Sigma Aldrich), followed by washing three times in TBST buffer. For AP development, membranes were incubated for 5–10 min at RT in the dark with 10 mL AP development buffer, containing 66 µL p-nitroblautetrazoliumchloride (NBT) and 33 µL 5-brom-4-chlor-3-indolylphosphate (BCIP). The reaction was stopped by washing the membranes in distilled water.

### 2.7. Flow Cytometric Analysis

For the analysis of transcriptional factor expression in CD4^+^ and CD8^+^ T cells, 1 × 10^6^ purified PBMCs were plated out per well in a 96-well conical bottom plate. For the intracellular staining of transcriptional factors, cells were fixed and permeabilized using the Fixation/Permeabilization buffer set (ThermoFisher/eBioscience). The following antibodies were used for the detection of cell surface and intracellular markers: CD4-PerCP-Cy5.5 (clone 0KT4), CD8-eF450 (clone OKT8), T-bet-PE-Cy7 (clone eBio4B10), GATA-3-eF660 (clone TWAJ), Foxp3-Alexa488 (clone 236A/E7), and RORγt-PE (clone AFKJS-9). In vitro expanded and restimulated PBMCs were stained using the same protocol and with the following markers for cell surface and intracellular marker detection: CD4-PerCP-Cy5.5 (clone 0KT4), CD8-eF450 (clone OKT8), IL-4-PE (clone 8D4-8), IL-10-PE (clone JES3-9D7), IL-13-APC (clone JES10-5A2), IL-17A-APC (clone eBio64DEC17), and IFNγ-eF660 (clone 4S.B3). Dead cells were excluded based on labeling with dead cell exclusion dye (Fixable Viability Dye eFluor780^TM^, ThermoFisher Scientific). Following intracellular staining, the cells were fixed in 200 µL 0.5% formaldehyde solution and were acquired on a BD Canto II (BD Biosciences). Antibodies were purchased from eBioscience (CD4, CD8, IL-17), ThermoFischer Scientific (IL-4, IFNγ, Fixable Viability Dye eFluor780^TM^), Biolegend (IL-10), and Miltenyi Biotec (IL-13).

### 2.8. In Vitro PBMC Expansion and Restimulation

Prior to in vitro expansion and restimulation, single cell suspensions were prelabeled with CFSE (diluted 1:1000 in 1 mL PBS) for 8 min at RT in the dark and were washed in 20 mL AB HS buffer (RPMI 1640, 5% human serum, 1% sodium pyruvate, 1% nonessential amino acids, 5% penicillin/streptomycin, 5% L- glutamine). CFSE-labeled cells were then plated out in a 96-well round-bottom plate at a concentration of 1 × 10^6^ cells per well in 200 µL complete medium (RPMI-1640, 5% human serum, 1% sodium pyruvate, 1% non-essential amino acids, 1% penicillin/streptomycin, 1% L-glutamine). PBMCs were expanded for 7 days at 37 °C, 5 % CO_2_ under the following conditions: unstimulated control (medium), αCD3 (2 µg/mL, clone Okt3), *T. suis* L1 stage larval antigen (TsL1, 5 µg/mL), *T. suis* crude adult antigen (TsAd, 20 µg/mL), *T. suis* adult excretory/secretory antigens (TsAdES, 20 µg/mL), MBP/MOG peptide mix (5 mM, Miltenyi Biotec). For restimulation on day 7, PBMCs were restimulated with PdBU (1:100,000) and Ionomycin (1:1000) for 30 min at 37 °C, 5% CO_2_, followed by another 3 h incubation with PdBU (1:100,000) and Brefeldin A (1:500) (ThermoFisher/Invitrogen).

### 2.9. Statistical Analysis

For assays and measurements, where an appropriate number of patient samples were available, statistical analysis was performed using GraphPad Prism Version 8 (San Diego, CA, USA). Results were tested for normal distribution using the Shapiro–Wilk normality tests, followed by an unpaired t-test or Mann–Whitney U test. Statistical significance is indicated as * *p* < 0.05, ** *p* < 0.01, *** *p* < 0.001.

## 3. Results

### 3.1. Immune Phenotyping of Peripheral Blood Mononuclear Cells (PBMC)

Here, we present the analysis of humoral and cellular responses of 11 individuals, namely six placebo-treated and five TSO-treated patients (Table 1). Within this cohort, patients had been diagnosed with MS 71 (±64) months prior to the start of the study. Due to the limited number of patients enrolled and individual patients not fully completing the clinical study visits, statistically significant group differences in clinical parameters were not expected. All primary and secondary endpoint parameters are summarized in Table 1 [16].

Mass cytometry was performed to assess the dynamics of peripheral blood immune cell subsets potentially induced by TSO treatment. Using an antibody panel of 43 cell-surface markers suitable to detect and characterize all major and many minor subsets of PBMC, we collected data from paired baseline and late study samples of four TSO and six placebo control patients, comprising an average of 170,000 CD45^+^ PBMC (min, 3600; max, 360,000 events). Pooled 43-dimensional data were subjected to dimension reduction by opt-SNE (Figure 1b), enabling the quantification of CD4^+^ and CD8^+^ T cell subsets, including regulatory, activated, MAIT and γδ T cells, B cells including naive and CD11c^+^ B cells, IgA^+^, IgM^+^, IgE^+^, and IgG^+^ memory B cells (the latter defined by the expression of CD27 and the absence of other cell-surface Ig). Additionally included in the analysis were IgG-, IgM-, and IgA-producing plasmablasts (PB), as well as basophils, monocyte subsets, plasmacytoid (pDC) and myeloid dendritic cells (mDC), and natural killer (NK) cells, based on their characteristic expression profiles (Figure 1b,c; Appendix A). We observed a mild expansion of pDC and mDC frequencies in several placebo and TSO-treated patients during the trial, however, no clear trend in DC frequencies was noted (data not shown). Similarly, assessment of basophil frequencies revealed mild increases in two placebo and three TSO-treated patients. Nevertheless, overall final-to-baseline basophil ratios in both patient groups were comparable, as were mean signal intensity (MSI) values for surface IgE on basophils (data not shown).

### 3.2. TSO-Treated Patients Display a Strong Parasite-Specific Antibody Response in Serum

To quantify and compare systemic antibody responses against *T. suis* in TSO-treated patients, antibody titers were measured via ELISA in serum samples collected at different time points during the trial. Our data show a strong, continuous increase over time of adult *T. suis* excretory-secretory (TsAdES) antigen-specific IgG and IgE levels in the TSO-treated group, paralleled by expectedly low and unchanged parasite-specific antibody levels in placebo controls (Figure 2a,b). Comparing IgG antibody subclass levels 12 months after treatment start, we also observed an induction of parasite-specific IgG1-4 levels in TSO-treated patients. All IgG subclasses were detected at higher levels compared with placebo-treated controls, with only one TSO patient displaying markedly lower IgG1 and IgG4 serum levels compared with the remaining individuals in this treatment group (Figure 2c). To assess the kinetics and diversity of antibody specificities to *T. suis* in TSO-treated patients, next we performed a Western blot to measure TsAdES-specific serum IgG. In TSO-treated patient 010, we could show the progressive recognition of a diverse set of *T. suis* protein bands recognized by serum IgG antibodies over the course of treatment (Figure 2d). More specifically, patient 010 displayed antibody recognition of a 17-18 kDa protein band at month 3 and a 16-17 kDa protein appearing later (month 6), as well as several bands ranging in size from 55 to 250 kDa from month 3 onwards (Figure 2d). At the final time point (month 12) of the trial we could also show that TSO-treated patients 002 and 013 displayed IgG protein recognition patterns distinct from patient 010 (Figure 2d), further strengthening the observation that TSO treatment of MS patients induces individualized antibody immune responses to the helminth.

Importantly, this diversity of protein recognition patterns developing over the course of the treatment shown by the immunoblot analysis coincides with the continuous increase in *T. suis*-specific serum IgG measured via ELISA. Overall, we therefore conclude that TSO-treated patients develop a robust parasite-specific antibody response to *T. suis*, indicated by increased titers of IgE and IgG antibody subclasses, in addition to recognizing a progressive diversity of parasite proteins over the course of 12 months of TSO treatment.

### 3.3. TSO Treatment is Associated with Mild Expansions of Circulating Memory B Cells and Plasmablasts

Mass cytometric analysis of additional cellular immune parameters showed that, while in placebo-treated patients naive B cell frequencies remained stable during the trial, TSO-treated individuals experienced a mild decrease in naive B cell frequencies, evidenced by declining final-to-baseline naive-to-total B cell ratios (Figure 3a,b). This observation was accompanied by a corresponding trend for elevated circulating IgM^+^IgD^−^ B cells in the TSO-treated group, compared with placebo-treated controls (Figure 3c). Moreover, we could show an increase in both IgA-switched and IgG-switched memory B cells in two out of four TSO-treated patients compared with the placebo-treated group, evident in the elevated final-to-baseline frequency ratios of IgA^+^ and IgG^+^ B cells (Figure 3d,e).

Assessment of HLA-DR^high^ plasmablast (PB) frequencies in the TRIOMS cohort also showed a trend for increased PB frequencies in the TSO-treated group, while two of four placebo-treated controls rather experienced a decline in circulating PB cells over time (Figure 3f,g). Consistent with the induction of *T. suis*-specific serum antibody levels in TSO-treated patients (Figure 2), mass cytometry further revealed notable dynamics in IgA^+^ and IgG^+^ PB populations in the TRIOMS patient cohort. Namely, while placebo-treated controls experienced an increase in IgA^+^ PB frequencies and a significant decrease in IgG^+^ PB during the study, TSO-treated individuals demonstrated a decrease in IgA-producing PB cells and a corresponding increase in IgG^+^ PB frequencies as a result of TSO treatment (Figure 3h,i). Overall, these results therefore confirm that experimental TSO treatment of MS patients induces a marked expansion of circulating antibody class-switched memory B cells, and the release of HLA-DR^high^ PB into the blood, characterized by an isotype shift in favor of IgG^+^ PB, consistent with the induction of parasite-specific IgG titers measured in the serum of TSO-treated patients.

### 3.4. MS Patients Display Variable T Cell Activation Profiles Following TSO Treatment

To characterize T cell-driven immune responses within the TRIOMS cohort, we analyzed the mass cytometry dataset for cellular phenotypes associated with T cell activation, as well as regulatory T cell (Treg) populations (Figure 4a). Assessment of HLA-DR^+^CD4^+^ activated T cell frequencies and final-to-baseline ratios could show that two out of four TSO-treated patients exhibited a notable increase in activated T cell frequencies following treatment, while in placebo-treated patients CD4^+^ T cell activation was less pronounced and even decreased during the study period (Figure 4b). A similar analysis of HLA-DR^+^CD8^+^ T cells revealed that in five out of six placebo-treated controls activated CD8^+^ T cell frequencies remained stable (Figure 4c). In contrast, however, the TSO-treated group showed a dichotomous response, with two TSO-treated patients displaying elevated CD8^+^ T cell frequencies and the other two showing markedly decreased CD8^+^ T cell activation in response to TSO treatment (Figure 4c).

Further analysis focused on Treg frequencies and activation in placebo and TSO-treated patients. While we did observe a comparable decline of total Treg frequencies in both patient groups during the study, final-to-baseline ratios indicated that this decrease appeared stronger in three out of four patients of the TSO-treated group and only two placebo-treated patients displayed a similarly drastic decline in total Treg frequencies (Figure 4d). Nevertheless, median signal intensity (MSI) of HLA-DR expression in Tregs revealed a trend for increased Treg activation in three out of four TSO-treated patients, while only one individual in the placebo group showed a similar increase in Treg activation (Figure 4e). In summary, using mass cytometric analysis of T cell subsets and activation phenotypes, we could show that TSO treatment of MS patients leads to diverse patterns of T cell activation. Namely, we observed that even though one TSO patient displayed a parallel increase in both CD4^+^ and CD8^+^ activated T cells (002), another patient lacked increased T cell activation at all (005), while the remaining two individuals displayed either increased CD4^+^ (008) or CD8^+^ T cell activation (010) only. Furthermore, these results also revealed that TSO treatment induces only mild activation of Tregs in MS patients and highlights a rather highly individual T cell response pattern of MS patients to experimental helminth treatment.

### 3.5. TSO Treatment Induces Variable T Helper Cell Responses in Individual MS Patients

Additionally, flow cytometry analysis was used to assess the kinetics of total CD4^+^ and CD8^+^ T cell frequencies and CD4^+^ T helper cell populations in TSO-treated versus placebo patients during the study. For this, PBMCs collected from each patient at different time points were analyzed for expression of the transcriptional factors defining the major T helper cell subsets, namely Tbet (Th1), Gata3 (Th2), Foxp3 (Treg), and RORγt (Th17) (Figure 5a). This assessment revealed that TSO-treated and placebo patients had stable and comparable total CD4^+^ and CD8^+^ frequencies for the entire duration of the study (Figure 5b). Calculations of CD4:CD8 T cell ratios (Figure 5c) confirmed this finding, fitting with the findings of the mass cytometry analysis described above (Figure 4).

Assessment of T helper cell subsets on an individual basis revealed a mild increase in Th2 cell frequencies in two TSO patients (patients 008 and 010) early after start of the treatment (Figure 5d). Notably however, these patients exhibited a contrasting Treg response, as patient 008 showed decreasing frequencies of Foxp3^+^ Treg, while patient 010 had a more pronounced Treg expansion during the study (Figure 5d). On the other hand, we observed no detectable changes in the frequencies of RORγt^+^ Th17 cells in neither placebo, nor in TSO patients during the study (Figure 5d). Th1:Th2 and Treg:Th17 ratios revealed diverse kinetics in the two patients groups over time. Two TSO patients (008 and 010) displayed decreasing Th1:Th2 ratios, while one placebo patient (011) had a marked increase by month 12 of treatment (Figure 5e). Treg:Th17 ratios appeared even more dynamic. There, two TSO-treated patients (008 and 013) displayed decreased Treg:Th17 ratios, in contrast to an increase seen in patient 010 (Figure 5e). Within the placebo group, two out of four individuals displayed a transient decrease, followed by a return to baseline values of Treg:Th17 ratios (006, 011) by month 12, while in the other two patients (003, 012) this ratio decreased by the end of the study. In conclusion, these results suggest that experimental helminth treatment of MS patients induces a variable T helper cell response, with notable differences between individuals.

### 3.6. MS Patients Display Limited Cytokine Recall Responses to Parasitic Antigens

We also investigated the in vitro cytokine recall responses to parasite antigens in both placebo and TSO-treated patients using PBMCs collected 12 months after start of the treatment (Figure 6a). Gating on proliferating CFSE^low^CD4^+^ T cells (Figure 6b,c), we observed similar CD4^+^ T cell proliferation in the four placebo patients and two TSO-treated patients in response to αCD3/CD28. CD4^+^ T cell proliferation was comparable, but lower, in response to different *T. suis* antigens and MBP/MOG antigen in both patient groups (Figure 6d). Notably, only placebo patient 012 displayed a markedly higher CD4^+^ T cell proliferation in response to parasite antigens (Figure 6d). Additionally, we measured cytokine production in proliferating CD4^+^ T cells in response to αCD3/CD28 stimulation, *T. suis* antigens and MBP/MOG antigen (Figure 6e,f). Interestingly, we observed notable frequencies of IL-10^+^CD4^+^ T cells in one of two TSO-treated patients in response to αCD3/CD28 and against all three *T. suis* antigens (TsL1, TsAd, and TsAdES). In contrast, an IL-10 recall response to *T. suis* antigens was largely absent in placebo patients, with only one placebo patient (011) showing an IL-10 recall response comparable to TSO-patient 010 following TsL1 stimulation (Figure 6e,f). For IL-13 and IL-17, placebo and TSO patients displayed similar recall responses to the different antigen stimulations. In contrast, two of four placebo patients showed a markedly stronger IFNγ recall response to TsL1 antigen, compared with the two TSO patients analyzed in this assay (Figure 6f). In summary, an overall comparable CD4^+^ T cell proliferation in response to parasite antigens was observed in both placebo and TSO-treated patients. Moreover, one of two TSO patients exhibited a stronger IL-10 recall response than placebo controls against several *T. suis* antigens, while IL-13, IL-17, and IFNγ recall responses to parasite antigens were rather comparable between the two patient groups.

## 4. Discussion

Helminth-based treatment and more specifically, the use of *Trichuris suis* eggs (TSO) to target various autoimmune diseases or inflammatory conditions have recently drawn considerable controversy. The inverse prevalence rates of chronic parasitic helminth infections and autoimmune diseases like multiple sclerosis (MS) across the world generally implies that the absence of naturally immunoregulatory active helminth infections in higher income communities and regions exacerbates the risk of autoimmune disease development and its clinical outcomes, including MS [9,27,28,29]. Nevertheless, robust clinical evidence of the efficacy of helminth treatment in interventional phase 2 studies is currently lacking [13,14].

MS is a multifaceted disease, comprising a range of disease subtypes [30,31], and with a variety of confounding factors involved in susceptibility, risk stratification, and treatment response capacities [32,33]. MS-centered studies therefore require strictly defined inclusion criteria, properly matched controls, and relatively large patient cohorts for clinical intervention studies. Although the TRIOMS study did not enroll a patient cohort large enough to fully evaluate treatment efficacy for the indicated clinical and radiographic endpoints, we employed a detailed exploratory immunological analysis at the individual patient level. Recently constructed genetic mapping of MS could identify the enrichment of MS susceptibility genes in various innate and adaptive immune cell populations, highlighting the potential contribution of functional shifts in one or more immune cell types to susceptibility to MS [33]. In the context of experimental treatments such as helminth-based therapy, however, a detailed survey of interindividual differences in parasite-specific immune responses at the cellular level had not been performed to date. For that purpose, we specifically looked into the individual cellular and humoral adaptive immune responses against several *T. suis* protein antigens. We combined global PBMC cell-surface marker expression-based immune phenotyping by mass cytometry with a more focused approach to address T helper cell subset differentiation, as well as parasite-specific antibody responses and T cell recall responses. A central objective was to delineate any stereotypical versus individual adaptive immune responses to TSO treatment in MS patients.

Our data suggest that the biweekly intake of 2500 TSO over 3–12 month consistently induces *T. suis* antigen-specific antibodies of the subclasses IgG1-4 and IgE in all patients receiving TSO treatment. Importantly, this is in line with other studies demonstrating distinct and diverse antibody responses to natural infection with gastrointestinal nematodes [34,35], as well as with clinical studies applying experimental helminth infections as treatment [13]. Moreover, for two TSO patients we observed a continuous elevation in *T. suis*-specific IgG and IgE levels during the 12-month study and in one of those patients we could show a corresponding diversifying parasite-specific IgG repertoire over time via immunoblotting. For any heterogeneous population, including the TRIOMS patient cohort, some degree of variability in humoral immune responses to any nematode is expected. Likewise, Haswell-Elkins reported on considerable heterogeneity in the antibody recognition profiles of *Ascaris lumbricoides* infected individuals in endemic regions, even when considering individuals from one household [36]. Overall, we could therefore demonstrate that MS patients mount normal parasite-specific antibody responses against *T. suis* as a result of prolonged TSO treatment over the course of 12 months.

Whether the continuous elevation of *T. suis* specific antibody titers in two patients can be explained by the cumulative antigen availability due to repetitive treatment, or potentially further larval development and hence antigen variation, cannot be conclusively answered, but has been demonstrated also by others [37]. It is likely that multiple rounds of exposure result in a continuous, but transient fluctuation of larval numbers in the patient’s gut and thus continuously increasing antibody titers might reflect increasing infection intensity over time as seen in natural *T. trichiura* infections [38]. Importantly, *T. suis* typically presents as a self-limiting infection in humans, with no indications of adult worms or ova developing during TSO therapy [39,40]. The antibody recognition analysis of one patient (patient 010) suggests predominantly a quantitative development of the antibody repertoire.

In a parallel approach, we employed a global PBMC immune phenotyping to study the cellular effects of TSO treatment on the basis of a baseline vs. post-treatment (final) comparison. Related to the increase of TSO-induced specific antibodies, our data show, on an individual level, the expansion of circulating IgG- and IgA-switched memory B cells, IgG^+^ plasmablasts, and HLA-DR^+^ plasmablasts, thus supporting TSO treatment effects on B cell responses. In turn, we observed no obvious baseline to final changes in the overall frequencies of natural killer (NK) cells, monocytes, basophils, dendritic cell (DC) subsets, γδ T cells, or mucosal-associated invariant T (MAIT) cells in either patient treatment group. 

Studying the T cell arm in the adaptive immune response of MS patients receiving TSO therapy is obviously hampered by disease-associated hyperactivity and metabolic alterations of T cells that result in highly individual and fluctuating T cell activation signatures in context of disease progression [41,42]. This is reflected when looking at T cells from a global perspective using multiparameter phenotyping. The baseline vs. final timepoint comparison of activated CD8^+^ cell frequencies varies greatly amongst individuals. However, our mass cytometry data revealed an increase of activated CD4^+^ T cells in two out of four patients receiving TSO, an effect that is less pronounced in the group of placebo-treated MS patients. Nevertheless, due to the small sample size, more substantial conclusions could not be made on the basis of these findings.

We performed further intranuclear transcription factor analysis to dissect the TSO-induced CD4^+^ T helper cell responses and their development over time. Again, we observed highly diverse kinetics of Tbet-expressing Th1 or GATA3^+^ Th2 cells, but a relatively stable population of RORγt^+^ Th17 cells within both the TSO- and the placebo-treated group. The induction of GATA3^+^ Th2 cells is a prototypical host immune response to helminth infections and the Th1/Th2 ratio is supposed to be a crucial point in assessing the effects of helminth therapy on Th1-driven inflammatory conditions. A recent study has addressed the heterogeneity of human Th2 cells during helminth infection in more detail [43]. They describe a decrease in a particular subset of CD27^−^CD161^+^ Th2 subset after deworming of helminth-infected individuals, which as a more pathogenic Th2 subset with enhanced effector functions potentially exert specific regulatory functions [43]. Their results therefore indicate that, in order to dissect the potentially balancing effects of Th2 versus Th1 responses under TSO therapy, an even more detailed analysis of both subsets would be required.

Until now, the precise role of Treg cells in relapses and remission in MS is not completely understood and a functional impairment of specific Treg subsets is thought to be a major driver of relapses in MS patients [44,45,46,47]. On the other hand, in the context of helminth treatment, the induction of Treg is the most prominent pathway for parasite-driven immunomodulation (reviewed in [48]). Our baseline vs. final comparison based on mass cytometry indicates no individual induction in Tregs for patients receiving TSO. Nevertheless, consistent with the overall CD4^+^ T cell activation, we observed an increase in activated Treg cells, as evidenced by their elevated HLA-DR expression. Further analysis of CD39- and CD73-expressing Treg subsets did not reveal additional hints for Treg subset regulation by TSO (data not shown). In line with the dynamics of Th1 and Th2 subsets during the study, transcription factor analysis of Foxp3^+^ CD4 T cells varied greatly over time in both patient groups, potentially reflecting the different clinical relapse courses of RRMS patients enrolled here.

In summary, the data presented here as part of a phase II pilot study of a limited cohort of MS patients receiving TSO as an alternative treatment approach (TRIOMS), indicates a high degree of interindividual variability in the adaptive immune responses to this helminth treatment. In light of the development of biologicals like TSO for the treatment of immune-mediated diseases like MS, future larger scale studies involving diverse MS patient cohorts should aim to better address and incorporate this interindividual variability of adaptive immune responses at the cellular level into any personalized medicine framework. For future studies focusing on the further development and optimization of helminth-based treatment for autoimmune diseases, the focus should therefore include a tailored treatment in order to meet the highly variable individual host cellular responses and to identify treatment “responder” and “nonresponder” patients, in particular in multifaceted diseases like MS. Biologicals like TSO might be highly beneficial for subgroups of MS patients, but to achieve this goal precision medicine probably is the choice to improve clinical treatment efficacy.

## Figures and Tables

**Figure 1 life-11-00101-f001:**
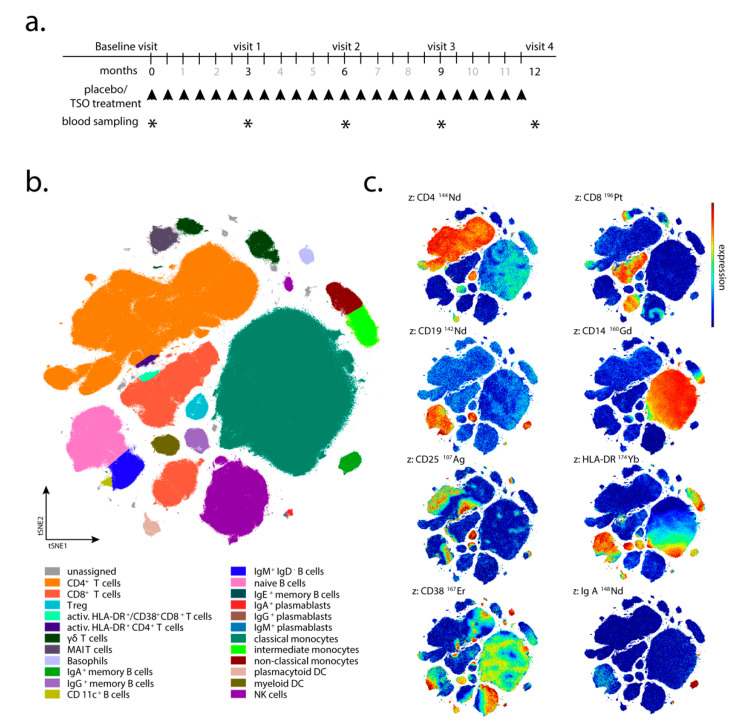
Study design. (**a**) Experimental design and patient treatment regimen (ClinicalTrials.gov: NCT01413243). At recruitment (baseline visit/month 0) patients started receiving either *Trichuris suis* ova (TSO) or placebo treatment every 2 weeks (twice monthly, indicated by black arrows). At 3, 6, 9, and 12 months post-start of the treatment (visits 1-4) blood sampling for peripheral blood mononuclear cell (PBMC) isolation was performed (indicated by asterisks). (**b**) PBMC analysis by mass cytometry. Global t-distributed stochastic neighbor embedding (tSNE) plot visualizing the major and minor clusters of blood immune cell subsets. Color-coded clusters were identified based on the expression levels of a 43-cell surface marker panel. (**c**) Exemplary t-distributed stochastic neighbor embedding (tSNE) plots visualizing the expression levels of 8 (out of the 43) markers defining distinct immune cell subsets within the global tSNE plot in (**b**), confirming the individually gated and identified clusters of “CD4^+^ T cells” and “CD8^+^ T cells” (CD4 and CD8 markers), B cell subsets (CD19 and IgA markers), “classical monocytes” (CD14 marker), “Treg” (CD25 marker) and activated immune cell subsets (HLA-DR and CD38 markers).

**Figure 2 life-11-00101-f002:**
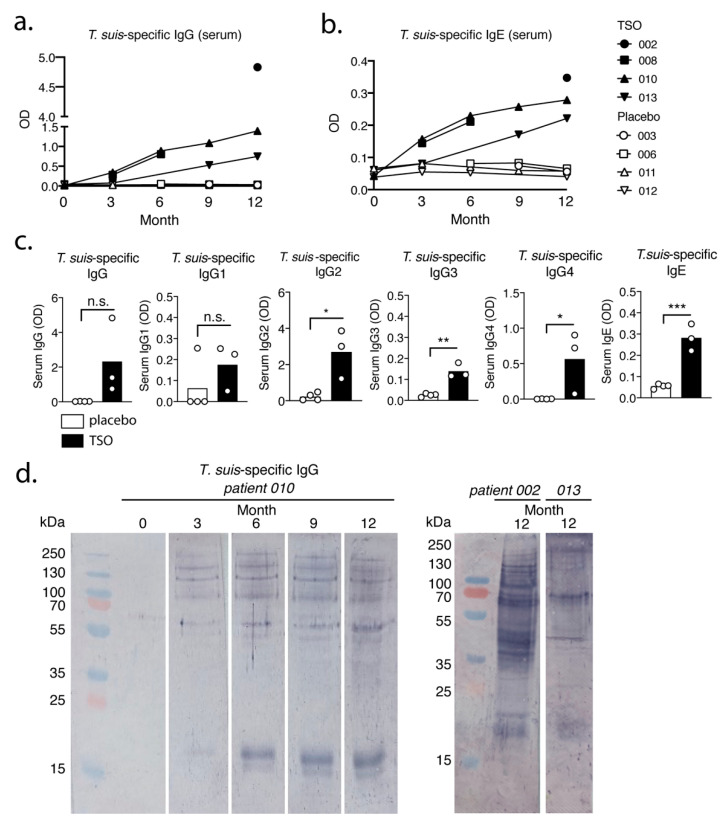
TSO-treated patients display significant parasite-specific antibody responses in serum against *T. suis*. Kinetics of *T. suis* adult excretory/secretory antigen (TsAdES)-specific (**a**) serum IgG and (**b**) serum IgE levels in placebo and TSO-treated patients. (**c**) Levels of TsAdES-specific IgG, IgG1, IgG2, IgG3, IgG4, and IgE 12 months following treatment start. (**d**) Immunoblot analysis displaying the kinetics and diversity of TsAdES-specific serum IgG for TSO-treated patient 010 (left panel) and for patients 002 and 013 at month 12 of treatment (right panel). In (**c**) results were tested for normal distribution using the Shapiro–Wilk normality test, followed by unpaired t-test or Mann–Whitney U test. * *p* < 0.05, ** *p* < 0.01, *** *p* < 0.001, n.s., not significant.

**Figure 3 life-11-00101-f003:**
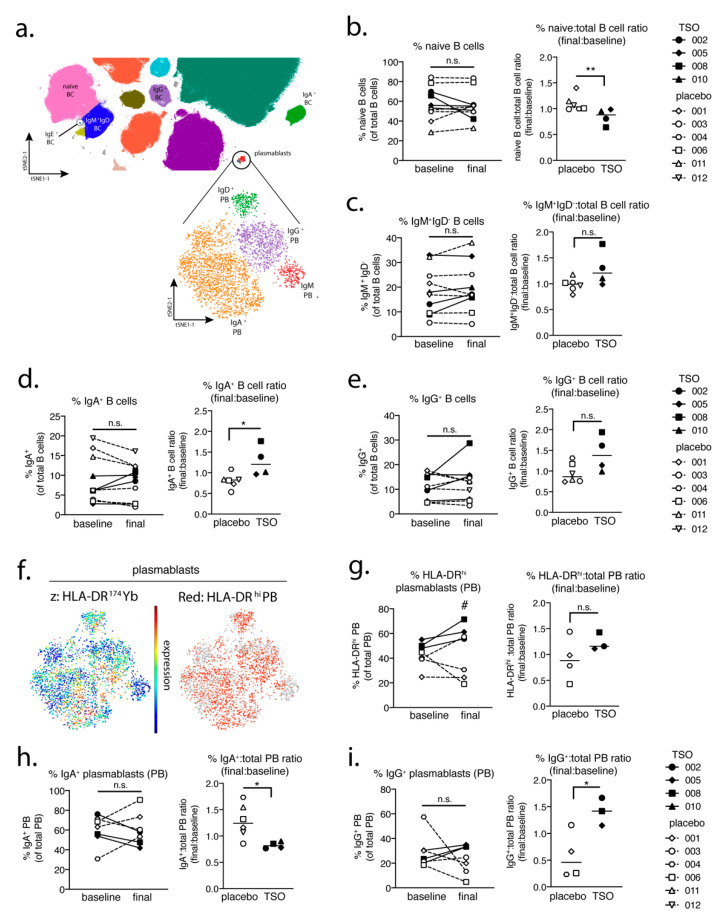
MS patients given TSO treatment display elevated frequencies of memory B cell subsets and a shift towards an IgG^+^ phenotype in the plasmablast (PB) compartment. (**a**) Extract of the tSNE plot from Figure 1; major B cell and PB populations are indicated. Frequencies of major B cell subsets (left) and ratios of these B cell subset frequencies among total B cells (final:baseline) (right) shown for (**b**) naive B cells, (**c**) IgM^+^IgD^−^ B cells, (**d**) IgA^+^ B cells, and (**e**) IgG^+^ B cells. (**f**) tSNE visualization of HLA-DR expression in the major plasmablast populations (lower panel in (a)). Plasmablast subset frequencies and final:baseline ratios of each subset among total plasmablasts for (**g**) HLA-DR^high^ PB, (**h**) IgA^+^ PB, and (**i**) IgG^+^ PB. The following time points were analyzed as “baseline” and “final” for each patient: months 0 and 12 (placebo patient 001), months 3 and 6 (placebo patient 003), months 0 and 9 (placebo patient 004), months 0 and 12 (placebo patient 006), months 0 and 12 placebo (patient 011), months 0 and 12 (placebo patient 012), months 0 and 12 (TSO patient 002), months 0 and 3 (TSO patient 005), months 0 and 3 (TSO patient 008) and months 0 and 12 (TSO patient 010). The calculated ratio values were tested for normal distribution using the Shapiro–Wilk normality test, followed by unpaired t-test or Mann–Whitney U test. *, *p* < 0.05; **, *p* < 0.01, n.s., not significant. In (g) # *p* < 0.05 comparing differences in the frequencies of HLA-DR^hi^ plasmablasts at the final timepoint between the TSO-treated patient group and placebo-treated group.

**Figure 4 life-11-00101-f004:**
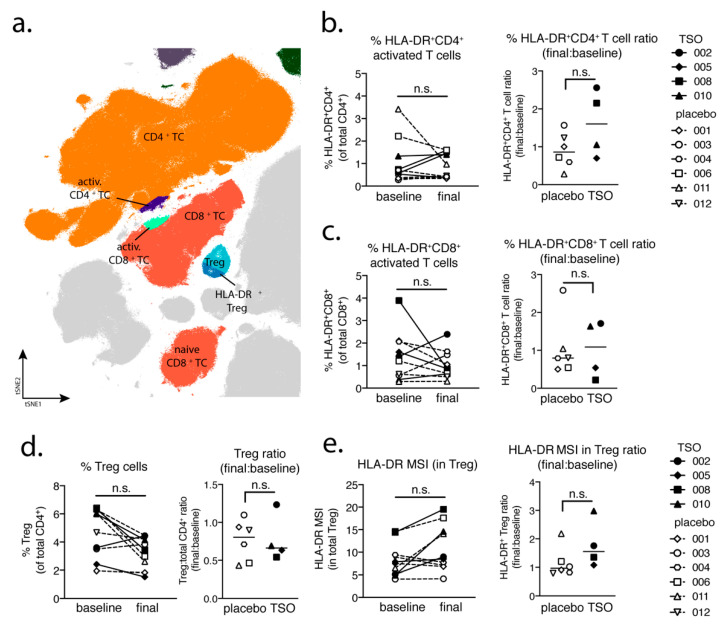
Individual patterns of T cell activation in TSO-treated patients. (**a**) Extract from the tSNE plot from Figure 1 illustrating the major CD4^+^ and CD8^+^ T cell populations analyzed in the TRIOMS patient cohort. T cell frequencies (left) and final:baseline ratios (right) for (**b**) HLA-DR^+^CD4^+^ T cells, (**c**) HLA-DR^+^CD8^+^ T cells, and (**d**) regulatory T cells (Treg). (**e**) Median signal intensity (MSI) of HLA-DR expression in total Treg cells (left) and HLA-DR mean signal intensity (MSI) ratios (final:baseline MSI values) (right). The following time points were analyzed as “baseline” and “final” for each patient: months 0 and 12 (placebo patient 001), months 3 and 6 (placebo patient 003), months 0 and 9 (placebo patient 004), months 0 and 12 (placebo patient 006), months 0 and 12 (placebo patient 011), months 0 and 12 (placebo patient 012), months 0 and 12 (TSO patient 002), months 0 and 3 (TSO patient 005), months 0 and 3 (TSO patient 008) and months 0 and 12 (TSO patient 010). The calculated ratio values were tested for normal distribution using the Shapiro–Wilk normality test, followed by unpaired t-test or Mann–Whitney U test. n.s., not significant.

**Figure 5 life-11-00101-f005:**
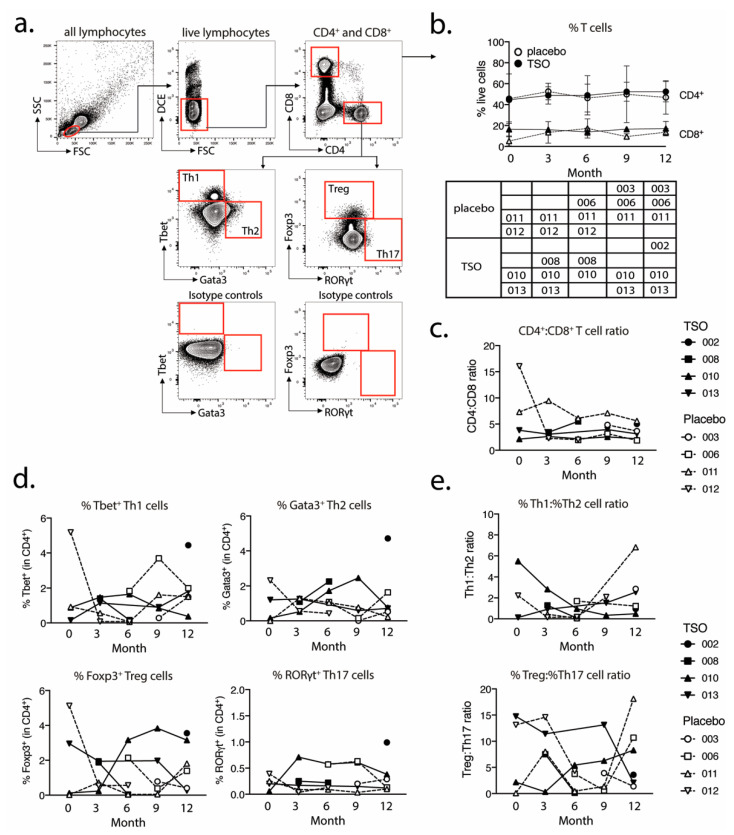
Placebo and TSO-treated patients display variable kinetics of circulating T helper cell subsets. (**a**) Gating strategy for flow cytometry analysis of Th1, Th2, Treg, and Th17 cell frequencies in PBMC samples of placebo and TSO-treated patients. Cells were initially gated on lymphocytes, based on their small size measured by forward (FSC) and side scatter (SSC) parameters. Within all lymphocytes, live cells were gated based on their negative expression of the dead cell exclusion dye (DCE). Within live lymphocytes CD4^+^ and CD8^+^ T cells were gated for further analysis. Within the CD4^+^ T helper cell gate, we defined the four major T helper cell subsets based on their expression of the transcription factors Tbet (Th1), Gata3 (Th2), Foxp3 (Treg), and RORγt (Th17), respectively. Respective isotype controls of the transcriptional factor stainings are depicted in the two lower plots. (**b**) Kinetics of total CD4^+^ and CD8^+^ T cell frequencies in placebo-treated and TSO-treated patients. The table under the graph indicates which patient samples were analyzed for each visit. (**c**) CD4:CD8 T cell ratio kinetics, based on the data in (b). (**d**) Kinetics of the frequencies of Tbet^+^ Th1 cells, Gata3^+^ Th2 cells, Foxp3^+^ regulatory T cells (Treg), and RORγt^+^ Th17 cells in placebo-treated and TSO-treated patients. (**e**) Ratios of Th1:Th2 and Treg:Th17 cells, based on the data in (**d**).

**Figure 6 life-11-00101-f006:**
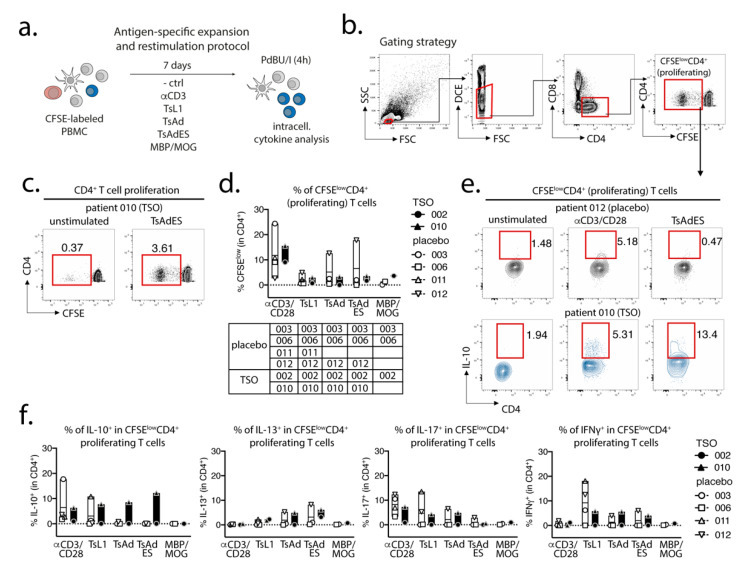
In vitro CD4^+^ T cell expansion and cytokine recall responses in placebo and TSO-treated patients. (**a**) Experimental set-up for CFSE-labeling, parasite-specific expansion, and unspecific re-stimulation of PBMCs from placebo and TSO-treated patient samples collected 12 months after the start of TSO treatment. (**b**) Gating strategy for analysis of proliferating CFSE^low^CD4^+^ cells. Cells were primarily gated on lymphocytes, based on their small size measured by forward (FSC) and side scatter (SSC) parameters. Within all lymphocytes, live cells were gated based on their negative expression of the dead cell exclusion dye (DCE). Within live lymphocytes, we gated on CD4^+^ T cells and further evaluated the CFSE^low^ subset of proliferating CD4^+^ T cells. (**c**) Exemplary dot plots of TSO-treated patient 010, showing CFSE^low^CD4^+^ T cell frequencies in an unstimulated versus a *T. suis* excretory-secretory (TsAdES)-stimulated sample. Frequencies of proliferating CFSE^low^CD4^+^ and cytokine-expressing T cells were corrected against background via subtracting frequencies detected in unstimulated controls for each patient sample and each restimulation condition (d-f). (**d**) Frequencies of proliferating CFSE^low^CD4^+^ T cells following in vitro expansion with αCD3/CD28 antibodies, *T. suis* L1 stage larval antigen (TsL1), *T. suis* crude adult antigen (TsAd), *T. suis* adult excretory/secretory antigens (TsAdES), and MBP/MOG peptides of placebo and TSO-treated patients. The table under the graph indicates patient sample availability for each stimulation condition. (**e**) Exemplary dot plots showing IL-10 production by CFSE^low^CD4^+^ T cells from placebo patient 012 and TSO-treated patient 010 following αCD3/CD28 and TsAdES restimulation, compared with respective unstimulated controls. (**f**) Frequencies of IL-10^+^, IL-13^+^, IL-17^+^, and IFNγ^+^ CFSE^low^CD4^+^ T cells of placebo- and TSO-treated patients after background subtraction.

**Table 1 life-11-00101-t001:** Demographic patient characteristics, baseline and endpoint clinical characteristics.

Individual Patient Demographic Characteristics										
Patient ID	Age (years)	Sex	Months since diagnosis	EDSS score (baseline)	EDSS score (after final treatment) ^#^	T2 lesions (baseline)	T2 lesions (after final treatment) ^#^	Gd enhancing lesions (baseline)	Gd enhancing lesions (after final treatment) ^#^	Number of relapses (V0-V4) ^#^
Placebo-treated patients										
001	53	M	24	2.5	3	48	50	0	0	1
003	42	M	76	0	0	50	53	1	0	0
004	36	F	128	2.5	2.5 (V2)	90	99 (V3)	NA	NA	2 (V2)
006	43	F	5	0	1	14	15	0	0 (V3)	0
011	56	F	53	4	2.5	NA	NA	NA	NA	1 (V2)
012	37	F	165	3.5	4	268	268	4	2	2
TSO-treated patients										
002	64	F	187	3.5	3.5	91	96	0	0	0
005	27	M	20	1	1 (V1)	37	39 (V2)	1	1 (V2)	2 (V1)
008	29	F	13	0	0 (V3)	107	105 (V3)	0	0 (V3)	0 (V3)
010	36	F	25	1.5	1.5	NA	NA	NA	NA	0
013	49	F	90	3	4	32	36	1	0	3

^#^ data from final visit (V4) if not indicated differently. NA: no data available.

## Data Availability

Mass cytometric data are available via www.flowrepository.org (FR-FCM-Z3F4). All other data presented in this work are available from the corresponding author upon reasonable request.

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
