# Peer review of "The Worm-Specific Immune Response in Multiple Sclerosis Patients Receiving Controlled Trichuris suis Ova Immunotherapy"

_life, 2021, doi:10.3390/life11020101_

Round 1

Reviewer 1 Report

Here Yordanova et al. describe the findings of a monocentric, prospective, randomized, placebo-controlled, double-blind, phase II pilot study of MS patients given repeating live T.suis or placebo. Analyses the PBMC and show that the parasite is detected by the immune system in infected patients. They perform CyTOF and multiparameter flow cytometry and find differences in immune subsets numbers and phenotype between cohorts.  

Strengths: Clinical trials require much time and effort, and the team is to be commended. The analysis is cutting edge.   

Weaknesses: The doses of T.suis would be difficult to standardise and MS is likely an umbrella term for dozens of disease subtypes (please discuss and cite genetic PMID:31604244  and microbiome PMID:32162850 confounders). Thus, statistical analysis of low patient numbers with inherent heterogeneity is challenging.

Comments:

Line 83: “the study showed that”

Line 128: Given the variability in CyTOF sensitivity with time, were the sample run in a single batch?

Line 183: version 8 should not be in brackets.

Line 199: “exams” should be “investigation”

Line 192: What was used as a standard curve?

Line 225: Thermofisher/eBioscience should be in brackets. The origin of the antibodies and fluorophores should also be stated. Were doublets gated out?

Line 257: “were not expected”

Line 258: “endpoint parameters are summarized”

Line 262: remove “systematically”

Line 279: Were the samples and placebos run in the same batch given the FACS laser changes strength (MFI) over time?

Line 291: “an expectedly”

General: An examination (Figure) of immune profiling with disease course/stage/outcome should be included.

Reviewer 2 Report

The manuscript  presents pilot study of a limited cohort  (5) of MS patients receiving TSO as an alternative treatment approach (TRIOMS). The results  indicate a very high degree of inter-individual variability in the adaptive immune responses to this treatment. However, the article is lacking details of biological and technical replicates. I would recommend having an external to read the paper, as there are many places where things are not well explained.

Major notes:

Figure 1. Study design. The relevance of the study in regards to experimental design is not clear. I do not find the Global t-distributed stochastic neighbor embedding (tSNE) plot visualizing clusters of blood immune cell subsets analysis particularly informative. What group does this data concern?

The Figure legend is not informative. Please describe the Fig 1 c and connection of the analysis with Fig1b.

Figure 2. The TSO-treated patients display parasite-specific IgG antibody responses in serum against T. suis. However, there is impossible to say that there is significant differences- but only a trend for changes-  as statistical analysis was not performed?. Why the results only for patients 2,8,19,13,3,6,11,12 are  present in the figure. In the figure legend should be explanation of the results obtained for patient 002. At the same figure we can find the information that (c) results were tested for normal distribution using the Shapiro-Wilk normality test, followed by unpaired t-test or Mann-Whitney U test. However there is not description of statistical analysis used in M&M section. 

According to these comments please verify Fig. 3 and Fig 4.

Figure 5. Please explain deeply the gating strategy. Please specified Why there is again 1 result (point of indication) for patient 002.

Figure 6 (f) The presentation of the results is unclear and the quality of figure should be improved.

Discussion. I would expect forward thinking statements and suggestions on the research results.

Could authors include some explanation of the result- very high degree of inter-individual variability in the adaptive immune responses to this TSO treatment. It seems to be important to verify the results of individual treatment with the baseline level of the regulatory T cells including their activity. Since the level of IL-13 was tested, was the level of IL-6 tested? IL-6 significantly influences the formation of the RORC / FOXP3-positive T cells population.

Minor notes:

M&M section:

Immunoblot analysis: Please specify which protein/ material were used for SDS-PAGE separation.

Tabel 1. Individual patient demographic characteristics. There is difficult to analyse information included in table 1. Authors need to divide the study groups into: placebo and TSA treated- instead number of samples with italic / no italic indication.

Round 2

Reviewer 2 Report

The revision has addressed my comments.